# Identification of key somatic oncogenic mutation based on a confounder-free causal inference model

Yijun Liu[1], Ji Sun[1], Huiyan Sun [1,2]*, Yi Chang[1,2]*

**1** School of Artificial Intelligence, Jilin University, Changchun, Jilin, China, **2** International Center of Future Science, Jilin University, Changchun, Jilin, China

* huiyansun@jlu.edu.cn (HS); yichang@jlu.edu.cn (YC)

## Abstract

Abnormal cell proliferation and epithelial-mesenchymal transition (EMT) are the essential events that induce cancer initiation and progression. A fundamental goal in cancer research is to develop an efficient method to detect mutational genes capable of driving cancer. Although several computational methods have been proposed to identify these key mutations, many of them focus on the association between genetic mutations and functional changes in relevant biological processes, but not their real causality. Causal effect inference provides a way to estimate the real induce effect of a certain mutation on vital biological processes of cancer initiation and progression, through addressing the confounder bias due to neutral mutations and unobserved latent variables. In this study, integrating genomic and transcriptomic data, we construct a novel causal inference model based on a deep variational autoencoder to identify key oncogenic somatic mutations. Applied to 10 cancer types, our method quantifies the causal effect of genetic mutations on cell proliferation and EMT by reducing both observed and unobserved confounding biases. The experimental results indicate that genes with higher mutation frequency do not necessarily mean they are more potent in inducing cancer and promoting cancer development. Moreover, our study fills a gap in the use of machine learning for causal inference to identify oncogenic mutations.

## Author summary

Identifying key mutations of cancers is helpful to better understand the mechanisms of cancer cell transformation and is critical for therapeutic approaches. Besides sequence and structure-based computational approaches, some functional impact-based methods which consider the association between mutation events and the activity of cancer-related biological processes have also been developed to detect key mutations. However, these methods mainly consider the correlation but ignore that the correlation is far from causality due to the existence of observed and unobserved confounding factors. We develop a confounder-free machine learning-based causal inference framework to estimate the causal effect of mutations on abnormal cell proliferation and epithelial-mesenchymal transition (EMT). It fills a gap in the use of causal mechanisms to discover potential driver

**Data Availability Statement:** All codes and data can be downloaded in GitHub: https://github.com/awa121/CauMu.

**Funding:** The authors thank funding support from the National Natural Science Foundation of China

(61902144 to HS; U19A2065 and 61976102 to YC). The funders had no role in study design, data collection and analysis, decision to publish, or preparation of the manuscript.

**Competing interests:** The authors have declared that no competing interests exist.

mutations in cancer biological systems. Applying our method to 10 cancer types, the identified key mutations are highly consistent with public well-verified ones. Additionally, some new key mutations have also been discovered.

This is a *PLOS Computational Biology* Methods paper.

## 1 Introduction

Understanding the mechanism of cancer initiation and development is of great significance for cancer treatment [1,2]. The initiation and development of cancer is a long-term, progressive multi-stage process, of which abnormal cell proliferation [3] and epithelial-mesenchymal transition (EMT) [4] are two important biological processes. Numerous studies have shown that the abnormal development of cancer cells is usually caused by gene mutations [5,6] and their abnormal regulation [7]. However, it is still not fully understood which mutations are the key ones and to what extent they affect different cancers in different stages.

Due to the complexity of cancer gene regulation and a large number of genes mutated in cancer, it is usually challenging to detect driver mutations by wet experiments. Various computational methods have been proposed to discover drivers and illustrate the mechanisms by which mutations regulate cancer development. Applying a large amount of multi-omics data [8], researchers usually identify cancer driver mutations by counting mutation frequency and performing a series of statistical analyses [9]: hypothesis testing [10], correlation analysis [11] and Bayesian statistics, machine learning, et.al. [12]. For example, DriverML [13] integrates Rao's score test [14] and supervised machine learning method to identify functional impacts of mutations. ActiveDriver [15] is proposed based on a generalized linear regression model to identify cancer driver genes through calculating mutation frequency in their protein signaling sites. DriverNet [16] formulates the estimation of the functional impact of mutations on mRNA expression networks as a minimum set cover problem and solves it by a greedy approximation algorithm. In addition to these generalized correlation-based approaches, several causal relationships characterizing mutations and phenotypic changes have also been proposed. For example, TieDIE [17] discovers the relationship between genomic perturbations and changes in cancer subtypes with a network diffusion approach to identify cancer driver modules. ResponseNet [18] captures the transcriptional changes in gene expression data to provide pathway-based causal explanations of disease genes. Although these methods aim at inferring the causality, they ignore the confounding bias due to the existence of confounders [19,20]. Moreover, the detection of key mutated genes is usually confounded by genomic heterogeneity, other mutations, and micro-environment stresses. Hence, it is natural to introduce causation theory and inference model to address this issue.

A fundamental challenge in causation studies is to eliminate confounding effects, especially when the data dimensions are extremely high. Confounders distort the relationship between treatment variables (e.g. mutation) and outcomes (e.g. cell proliferation), leading to erroneous results. For example, when estimating the causal effect of the mutated *TP53* on cell proliferation, oxidative stress may be a confounding factor as it affects both the mutation probability of *TP53* [21,22] and cell proliferation degree [23]. It leads to bias the true effect of *TP53* mutation on cell proliferation when the distribution of oxidative stress levels is different between *TP53* mutated and non-mutated sample groups [24]. Traditional statistical causal models reduce the influence of confounders by balancing confounding variables across groups [25],

standardizing and stratifying data [26,27], or performing regression analysis between confounders and treatment variables on observational data [27]. But these causal models are based on the assumption of unconfoundedness, i.e., all confounders are observable, which is too strong in many complex biological system studies. For instance, we can neither exactly know what causes mutations nor measure most micro-environment stresses. Stronger assumptions usually simplify operations but deviate from reality, so researchers need to make a trade-off between the reality or credibility of assumptions and the precision of an identifiable result [28]. In addition to unobserved confounders, there are still other challenges to the causal effect model in biological studies, such as the high dimension of data and the non-linear relationship between variables.

In this paper, to relax the assumption of unconfoundedness, we consider both observed and unobserved confounders in the causality model for estimating the causal effect of somatic mutations on cell proliferation and EMT processes which are two signature biological processes of cancer. We begin with a structural causal model to intuitively show causal relationships between the variables in the biological system. Then, we propose a framework to estimate the Causal Effect of a mutation on cancer Biological Process (CEBP), where mutations with larger effects are key mutations. More specifically, we formulate activities of the specific biological process as the outcome and address the causal inference problem with a generative model, which learns the representation of unobserved confounders. We apply the proposed framework to predict key mutations of 10 cancer types through the genomics and transcriptomics data from The Cancer Genome Atlas (TCGA) database. Most key mutations identified by CEBP are highly consistent with the existing literature and experimental findings. Beyond that, we calculate the pure effect of the top 10 genes with the highest mutation rates for each cancer type. The results show that there is no perfect positive correlation between genes' mutation rates and causal effects on cancer cell abnormal proliferation and EMT processes. Besides, CEBP is a general framework and can be easily extended to other studies for discovering key mutations in other important biological processes of cancers and even other diseases.

## 2 Results

### 2.1 The structural causal model and analysis pipeline of CEBP

For the study of the causal relationship between cancer biological processes and mutations, we construct the Structural Causal Model (see Fig 1) where the nodes represent the variables and the arrows indicate the direction of causality. The outcome of the causal system denoted by $Y$ is the biological processes activity of samples, the binary treatment variable denoted by $M$ is the somatic mutation data of a gene (denoted by $g$), and the observed confounders denoted by $X$ is somatic mutation matrix made up of other genes except g. Although we cannot directly take action on the unobserved/hidden confounders $Z$, it is possible to find proxies [29] for them and recover the posterior distribution from the observations (X, M, Y) by the generative model, such as variational autoencoder [30]. One crucial step in inferring the causal relationship ($M{\rightarrow}Y$) is to eliminate the misleading effect caused by confounders which affect both treatment ($M$, via $Z{\rightarrow}M$) and outcome ($Y$, via $Z{\rightarrow}Y$) and lead to the spurious statistical correlation between $M$ and $Y$ [31]. Specifically, the inputs are the somatic mutation matrix of $K$ genes and a binary mutation vector $g$ for $N$ samples, the output is the causal effect of mutation $g$ on a cancer biological process:

- $M = (m_1,\ldots,m_N)^T \in \mathbb{R}^N$ with $m_i = 0/1$ represents the gene $g$ mutated or not in i-th sample.

- $Y = (y_1,\ldots,y_N)^T \in \mathbb{R}^N$ is the vector of biological process activity (cell proliferation and EMT process, in this paper) for cancer samples.

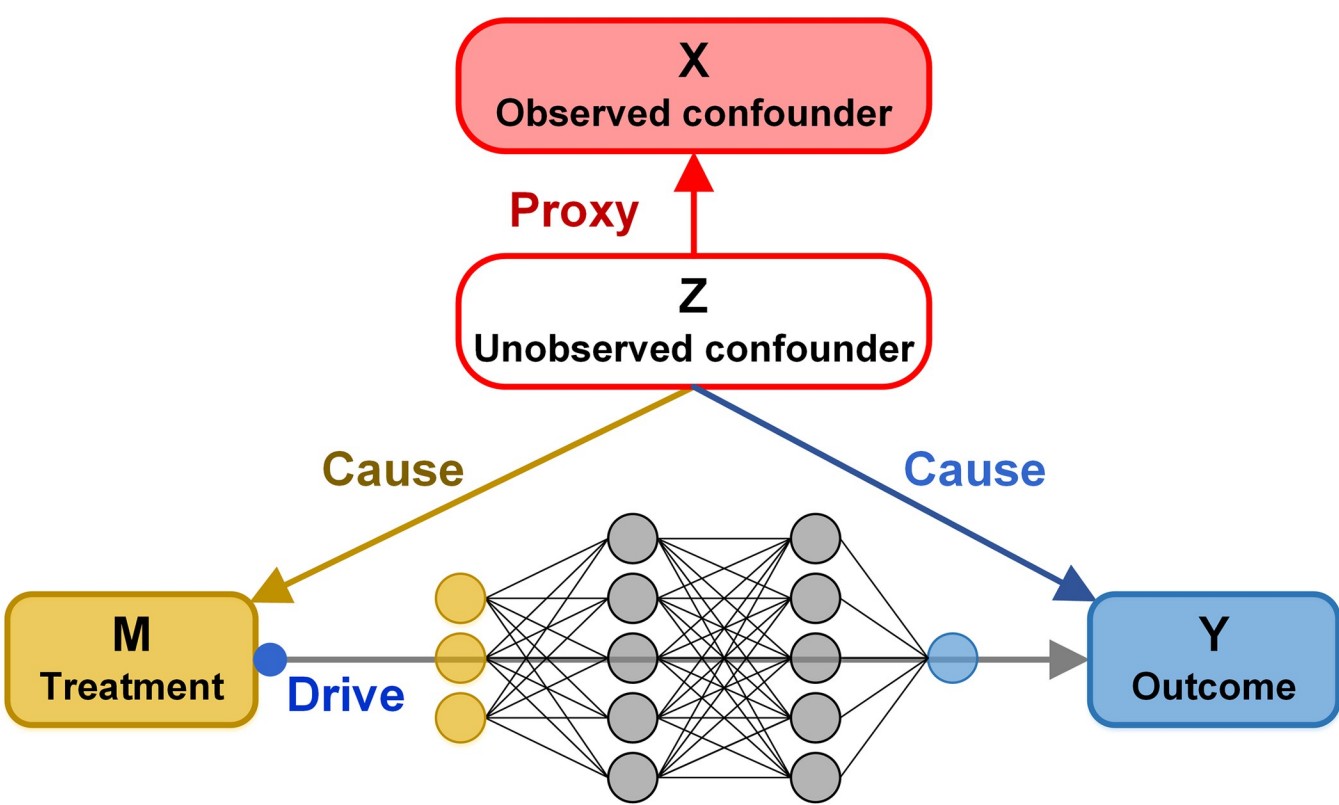

**Fig 1. The causal structure of the mutation in the cancer biological process.** *M* is a binary treatment variable, i.e., whether a gene mutated or not; *Y* is the outcome, e.g., the cell proliferation activity. *X* is the observed confounders, i.e., the mutation matrix of covariant genes; *Z* is the unobserved confounders which can be learned from *X*, e.g., micro-environment stress, so another role for *X* is a proxy variable for *Z*; Confounder affect both the treatment and the outcome and may lead to the erroneous association.

- $X = (x_1, \ldots, x_N)^T \in \mathbb{R}^{N \times K}$ is the somatic mutation matrix where $x_i = (x_i^1, \ldots, x_i^K)^T \in \mathbb{R}^K$ with $x_i^j = 0/1$ representing whether the j-th gene of i-th sample mutated or not.

- *Z* is the hidden confounder of the causal system. As some causes of the development of cell replication and EMT are hard to measure and cannot be directly observed, we sum it up as hidden confounder *Z*.

- *Z*→*X*: Another role of the observed confounder *X* is considered as a proxy variable for hidden confounder *Z*, which can be learned from *X*.

- *Z*→*M*: Z affects mutation of *g*.

- (Z, M)→*Y*: The edges show that Y is affected by two paths: (1) the direct path *M*→*Y* which denotes causal effect of mutation *g* on the cancer biological process, and (2) the path *Z*→*Y* indicates that the outcome is affected by the hidden confounder *Z* (e.g. micro-environment stress) and observed confounder *X* through *Z*.

The goal of this paper is to recover the causal effect of a mutation on a cancer biological process, that is to recover the average treatment effect (ATE) of the treatment on the outcome from observations (X, M, Y):

$$ATE := \mathbb{E}[Y_i(m=1) - Y_i(m=0)] = \frac{1}{N}\sum\nolimits_{i=1}[Y_i(m=1) - Y_i(m=0)], \tag{1}$$

where $Y_i(m=1)$ and $Y_i(m=0)$ denote the outcome of the i-th sample with and without being treated (the biological process activity of the i-th sample with and without gene mutated), respectively. The positive ATE value means that the mutation promotes this biological process, zero and negative ATE values mean the mutation has no effect and does not promote this biological process. Although we cannot simultaneously observe both $Y_i(m=1)$ and $Y_i(m=0)$ for the i-th sample, we can recover the counterfactual outcome (the one that cannot be observed is known as the counterfactual outcome) by capturing the structure of latent variables through the variational autoencoders (VAEs) model and calculate ATE. So we proposed a framework called CEBP to estimate ATE.

CEBP consists of two parts: estimation of the cancer biological process activity in each sample based on regression analysis on core features, as no direct experimental data are available for the outcome of the causal model; and the variational autoencoder model to predict the causal effect (see Fig 2). Specifically, for a biological pathway with $P$ genes, we first generate the transcriptomics data matrix of $N$ samples on the pathway and calculate the Person correlation coefficient matrix between $P$ genes. Then, we select a set of core genes which are highly correlated with each other. Finally, we use the linear regression coefficient between core genes' expression value of each sample and core genes' average expression value vector on $N$ samples, as the biological process activity of each sample (i.e. outcomes of the model). Theoretical details of the calculation are given in Section 3.2.

With the output of core feature regression, we get the observations $\{(x,m,y)\}$ and use a deep latent-variable model called Causal Effect Variational Autoencoder (CEVAE)[30] for causal inference. This method is a generative model consisting of an inference network as the encoder and a model network as the decoder. The inference network learns a multivariate Gaussian distribution $\mathcal{N}(\mu_z, \sum_z)$ from which we sample the latent variable $Z$, and the model network reconstructs the data $\{(\bar{x}, \bar{m}, \bar{y})\}$ from the prior distribution of $Z$. By training to minimize the KL divergence between real data and reconstruction, the optimized CEBP averages the outcomes across different groups eliminating confounders and estimates ATE of the mutation on the biological process. The details of the calculation are given in Section 3.3.

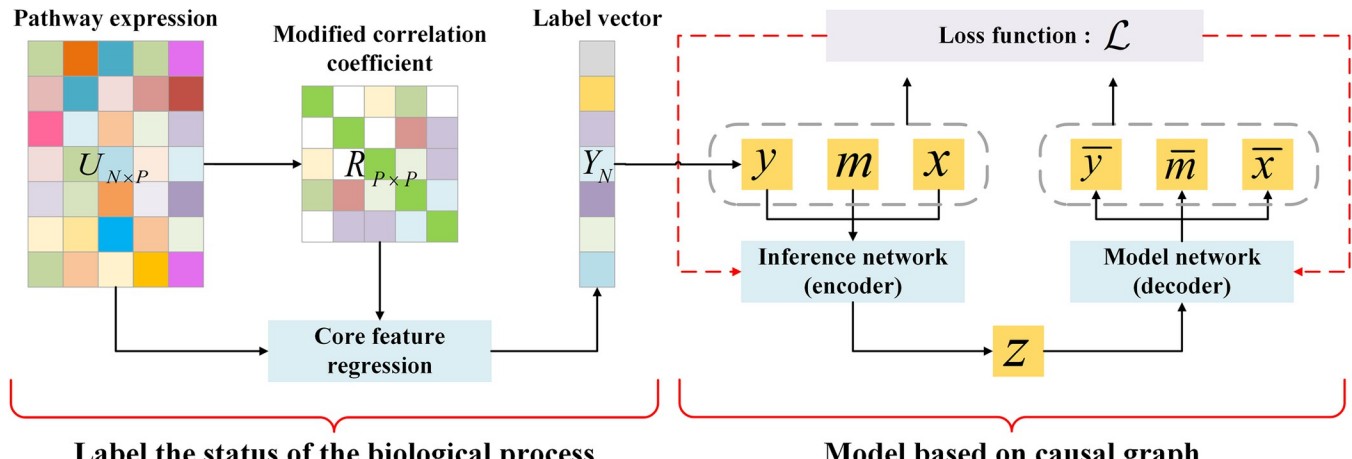

**Fig 2. An illustration of the proposed framework.** The biological process activity of each sample is obtained by core genes regression and used as the outcome of the casual model. The casual model consists of an inference network to calculate the mean and variance of $P(Z|X,y,m)$ and a model network to recover $P(Z,X,y,m)$.

## 2.2 Identification of key mutations leading to DNA replication and EMT

We apply CEBP to identify key mutations that lead to DNA replication and EMT process for 10 cancers from some candidate mutations. As the number of samples and the number of genes with high mutation rates vary between cancers, we set different search cutoff of key mutations for different cancers: we identify key mutations of BRCA, KIRC, HNSC, LIHC, ESCA, and KIRP from the genes with top 200 high mutation rates; we discover key mutations of LUAD, LUSC, and BLCA from the genes with mutation rates above 7%, and THCA from the genes with mutation rates above 1%. When setting the dimensions of confounding variables for CEBP, we select the top 200 genes with the highest mutation frequency as the observed confounders, and set the dimensions of unobserved confounders as 20 for identifying the key mutations of both DNA replication and EMT process.

We rank the ATE of candidate key mutations and select the top 10 mutations with highest ATE values, as shown in Fig 3 for DNA replication and Fig 4 for EMT process. We assume that the mutation with a higher ATE is more likely to be a key one. Most key mutations selected by CEBP have been reported or proved to play important roles in cancer-related biological processes in existing studies. For example, *RB1* [32,33] and *IGSF10* [34] in BRCA have been proved associated with DNA replication. The *BRAF* encodes a serine/threonine protein kinase, which is involved in the regulation of transcriptional activity during cell growth, division and differentiation, and its mutation is associated with the development of the thyroid tumor [35]. *NFE2L2* is a transcription factor that primarily affects the expression of

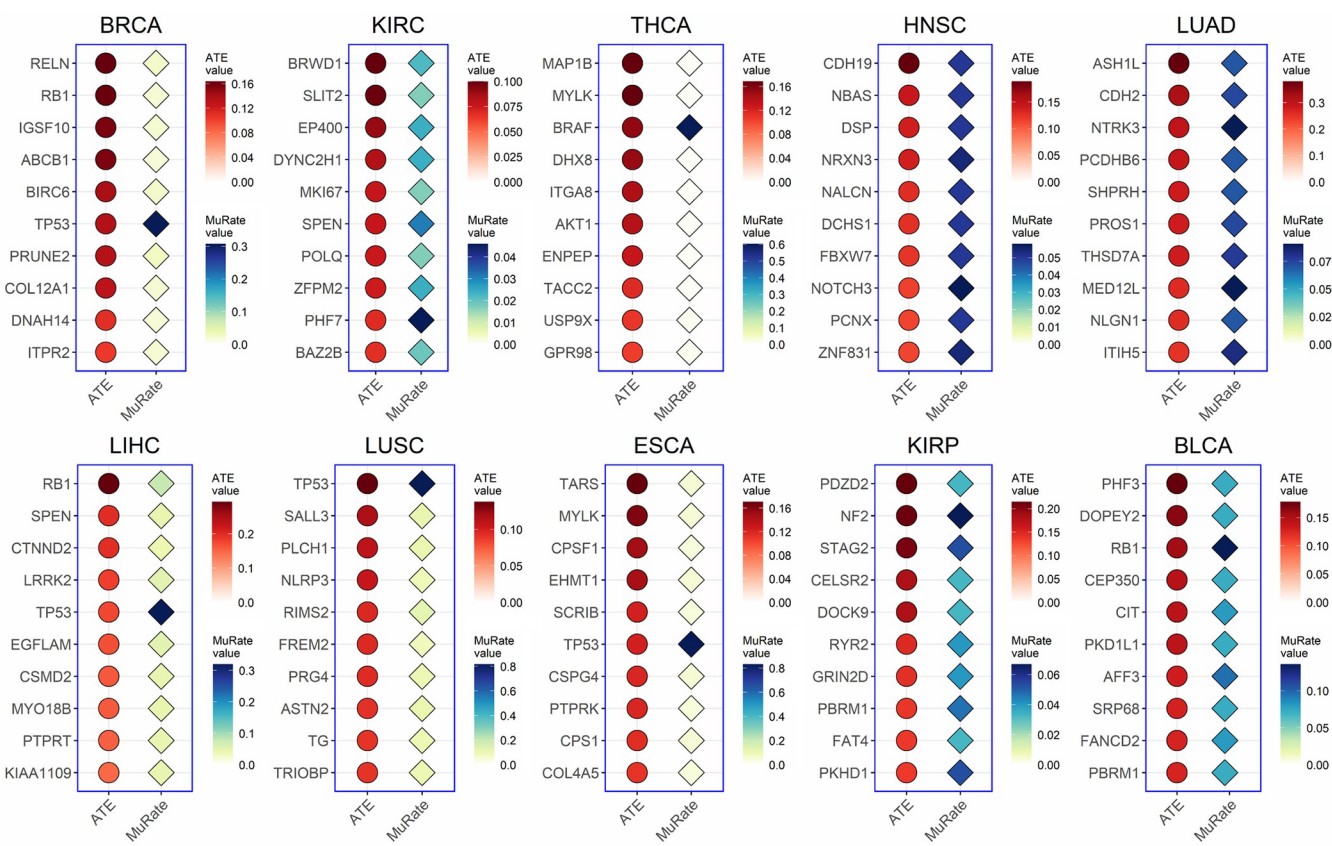

**Fig 3. Top 10 mutations with the highest causal effect on DNA replication in 10 cancers.** The figure shows the ATE value and the mutation rate of each predicted key mutation.

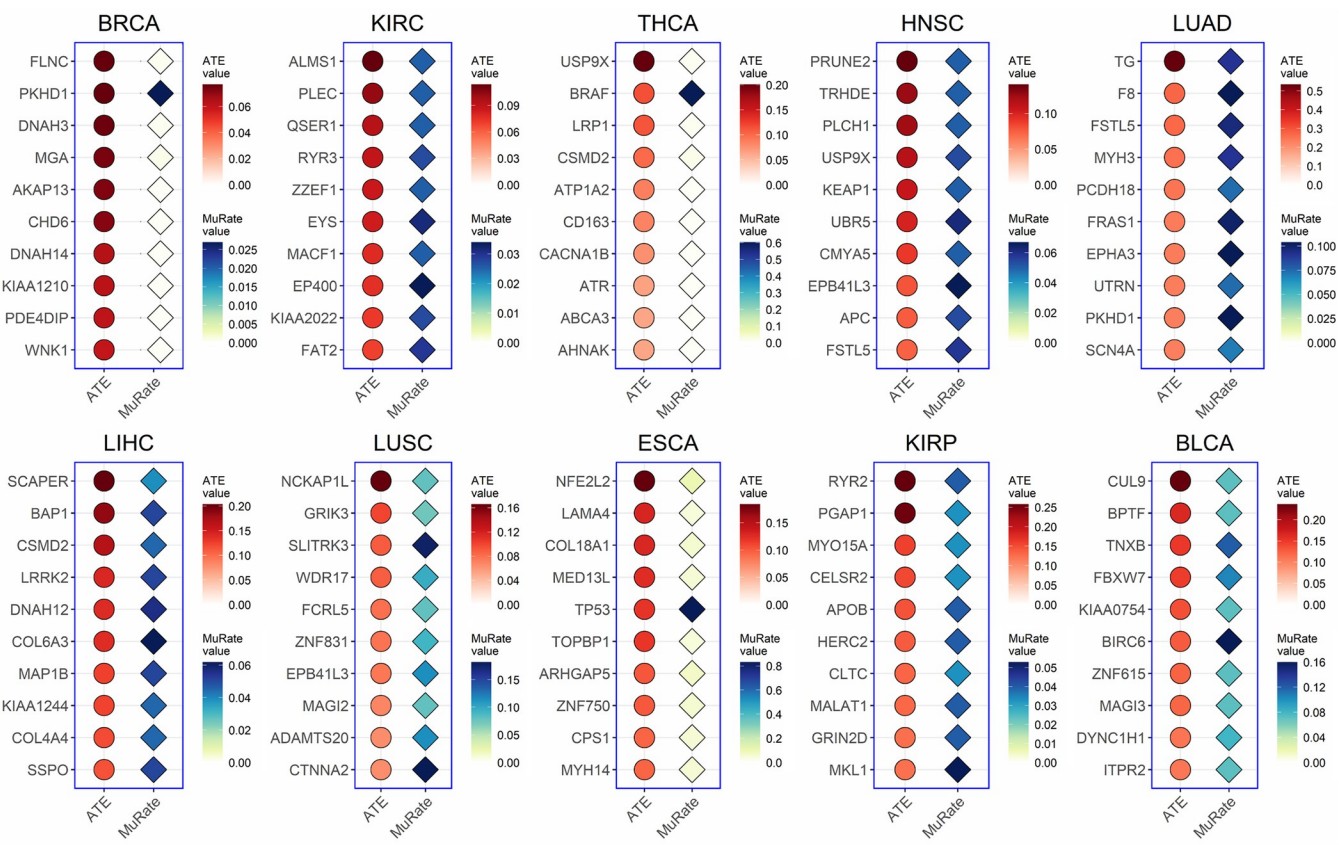

**Fig 4. Top 10 mutations with the highest causal effect on EMT process in 10 cancers.** The figure shows the ATE value and the mutation rate of each predicted key mutation.

antioxidant genes and its activation regulates a variety of cancer hallmarks related to EMT, including tumor aggressiveness, invasion, and metastasis formation [36], and has a significant impact in esophageal carcinoma (ESCA) [37].

Moreover, Figs 3 and 4 shows that some genes with low mutation rates have high ATE on DNA replication and EMT, while previous studies tend to pay much attention on genes with high mutation rates. Our results suggest that there is no obvious correlation between the mutation rates and their impact on the activity of cancer-related biological processes, and mutations with high ATE values are in need of more attention.

## 2.3 Causal effect of mutations with the highest mutation rates on DNA replication and EMT

We check the causal effect of mutations with high mutation rates on cell proliferation and EMT processes in 10 cancers. We select top 10 mutations with the highest mutation rates in each cancer and calculate their causal effect (ATE) on DNA replication and EMT process. The mean and standard deviation of 10 independent runs are shown in Table 1 and Table 2, the genes are listed in descending order of mutation rates from left to right, and the gene with the highest ATE for each cancer is highlighted.

Genes with higher mutation frequency do not necessarily indicate they have higher causal effects on cancer biological processes. For example, *TTN* has the highest mutation frequency in LUAD, but rare relevant research or wet experiment indicates that its mutation is associated

**Table 1. The causal effect of the top 10 genes with highest mutation rates on DNA replication.**

| Cancer type | Gene: ATE, Genes from high mutation rate to low | | | | | | | | | |
|---|---|---|---|---|---|---|---|---|---|---|
| BRCA | PIK3CA:-0.055 ±0.003 | **TP53:0.14 ±0.004** | TTN:0.015 ±0.003 | CDH1:-0.099 ±0.002 | MUC16:0.003 ±0.005 | GATA3:-0.027 ±0.005 | MAP3K1:-0.078 ±0.006 | MUC4:0.03 ±0.008 | RYR2:0.027 ±0.006 | MUC5B:0.008 ±0.005 |
| KIRC | VHL:0.009 ±0.004 | PBRM1:-0.029 ±0.003 | MUC4:0.024 ±0.004 | TTN:-0.006 ±0.006 | MUC16:0.027 ±0.007 | MTOR:0.012 ±0.006 | KDM5C:0.007 ±0.006 | | DST:0.016 ±0.013 | AHNAK2:-0.04 ±0.009 |
| THCA | **BRAF:0.149 ±0.004** | NRAS:-0.1 ±0.007 | MUC16:0.045 ±0.009 | TTN:0.055 ±0.008 | **SETD2:0.043 ±0.005** | HRAS:-0.004 ±0.012 | DNAH9:0.032 ±0.012 | ZFHX3:0.023 ±0.013 | CSMD2:0.085 ±0.024 | OTUD4:-0.052 ±0.011 |
| HNSC | TP53:-0.079 ±0.01 | TTN:-0.01 ±0.005 | FAT1:-0.072 ±0.008 | LRP1B:0.035 ±0.009 | MUC16:0.01 ±0.008 | CDKN2A:-0.067 ±0.006 | PIK3CA:-0.023 ±0.01 | NOTCH1:-0.05 ±0.007 | SYNE1:-0.021 ±0.014 | **PCLO:0.101 ±0.007** |
| LUAD | TTN:0.006 ±0.008 | **TP53:0.132 ±0.01** | MUC16:0.001 ±0.007 | RYR2:0.008 ±0.009 | LRP1B:0.002 ±0.011 | KRAS:-0.11 ±0.01 | ZFHX4:0.016 ±0.012 | FLG:0.04 ±0.008 | MUC17:0.056 ±0.008 | FAT3:0.019 ±0.014 |
| LIHC | TTN:-0.014 ±0.012 | **TP53:0.188 ±0.007** | CTNNB1:-0.1 ±0.009 | APOB:-0.047 ±0.01 | RYR2:0.03 ±0.008 | OBSCN:0.003 ±0.009 | USH2A:-0.0 ±0.023 | ABCA13:-0.007 ±0.017 | CSMD1:0.044 ±0.017 | GPR98:0.095 ±0.016 |
| LUSC | **TP53:0.129 ±0.012** | TTN:0.037 ±0.01 | MUC16:0.048 ±0.007 | RYR2:-0.006 ±0.007 | ZFHX4:-0.04 ±0.006 | LRP1B:-0.005 ±0.005 | SYNE1:0.041 ±0.011 | RYR3:0.02 ±0.011 | NAV3:0.007 ±0.01 | PKHD1L1:0.038 ±0.006 |
| ESCA | **TP53:0.108 ±0.009** | DST:0.031 ±0.01 | MUC4:-0.014 ±0.01 | MACF1:0.003 ±0.012 | RNF213:0.05 ±0.011 | MUC5B:-0.032 ±0.011 | GNAS:-0.045 ±0.016 | MALAT1:0.034 ±0.007 | MUC17:-0.023 ±0.021 | SYNE2:-0.004 ±0.009 |
| KIRP | FRG1B:-0.009 ±0.008 | **TTN:0.056 ±0.007** | MUC2:0.024 ±0.009 | MUC4:-0.026 ±0.017 | MUC5B:0.037 ±0.006 | NEFH:-0.054 ±0.008 | FAT1:0.023 ±0.011 | MAML2:-0.029 ±0.018 | MUC16:0.041 ±0.012 | OBSCN:-0.05 ±0.007 |
| BLCA | TTN:0.027 ±0.012 | **TP53:0.112 ±0.012** | MUC16:0.017 ±0.009 | ARID1A:0.011 ±0.009 | KDM6A:-0.022 ±0.01 | SYNE1:0.015 ±0.012 | PIK3CA:-0.001 ±0.016 | HMCN1:-0.02 ±0.013 | FLG:0.061 ±0.013 | CSMD3:-0.022 ±0.015 |

**Table 2. The causal effect of the top 10 genes with highest mutation rates on EMT.**

| Cancer type | Gene: ATE, Genes from high mutation rate to low | | | | | | | | | |
|---|---|---|---|---|---|---|---|---|---|---|
| BRCA | PIK3CA:0.022 ±0.003 | **TP53:0.048 ±0.002** | TTN:-0.029 ±0.003 | CDH1:0.036 ±0.005 | MUC16:-0.043 ±0.004 | GATA3:-0.062 ±0.006 | MAP3K1:0.016 ±0.004 | MUC4:-0.029 ±0.005 | RYR2:-0.053 ±0.012 | MUC5B:-0.024 ±0.007 |
| KIRC | VHL:-0.012 ±0.004 | PBRM1:-0.026 ±0.004 | MUC4:-0.029 ±0.007 | TTN:0.01 ±0.006 | SETD2:-0.0 ±0.004 | **MUC16:0.031 ±0.008** | MTOR:-0.025 ±0.008 | KDM5C:-0.026 ±0.006 | DST:-0.032 ±0.013 | AHNAK2:-0.04 ±0.009 |
| THCA | **BRAF:0.103 ±0.006** | NRAS:-0.168 ±0.006 | MUC16:-0.018 ±0.012 | TTN:-0.011 ±0.014 | TG:-0.054 ±0.005 | HRAS:-0.105 ±0.01 | DNAH9:-0.13 ±0.015 | ZFHX3:0.041 ±0.013 | CSMD2:0.09 ±0.008 | OTUD4:0.054 ±0.008 |
| HNSC | **TP53:0.064 ±0.008** | TTN:0.03 ±0.005 | FAT1:0.017 ±0.009 | LRP1B:-0.047 ±0.006 | MUC16:-0.004 ±0.008 | CDKN2A:-0.011 ±0.009 | PIK3CA:-0.021 ±0.01 | NOTCH1:-0.003 ±0.008 | SYNE1:-0.05 ±0.008 | PCLO:-0.005 ±0.007 |
| LUAD | **TTN:0.042 ±0.005** | TP53:0.027 ±0.008 | MUC16:0.009 ±0.007 | RYR2:-0.033 ±0.007 | LRP1B:-0.009 ±0.01 | KRAS:0.006 ±0.006 | ZFHX4:-0.003 ±0.006 | FLG:-0.016 ±0.008 | MUC17:-0.06 ±0.01 | FAT3:-0.004 ±0.009 |
| LIHC | TTN:-0.053 ±0.007 | TP53:-0.011 ±0.008 | CTNNB1:-0.128 ±0.011 | APOB:0.005 ±0.009 | **RYR2:0.029 ±0.011** | OBSCN:0.021 ±0.014 | USH2A:0.006 ±0.018 | ABCA13:-0.016 ±0.007 | CSMD1:-0.025 ±0.016 | GPR98:0.034 ±0.015 |
| LUSC | TP53:-0.045 ±0.014 | TTN:-0.001 ±0.006 | MUC16:-0.022 ±0.009 | RYR2:-0.021 ±0.006 | ZFHX4:0.014 ±0.009 | LRP1B:-0.008 ±0.011 | **SYNE1:0.032 ±0.011** | RYR3:0.006 ±0.008 | NAV3:-0.008 ±0.012 | PKHD1L1:0.017 ±0.011 |
| ESCA | **TP53:0.103 ±0.015** | DST:0.045 ±0.011 | MUC4:0.054 ±0.011 | MACF1:-0.1 ±0.01 | RNF213:0.002 ±0.008 | MUC5B:0.056 ±0.014 | GNAS:-0.078 ±0.01 | MALAT1:0.025 ±0.009 | MUC17:-0.038 ±0.006 | SYNE2:0.04 ±0.014 |
| KIRP | FRG1B:-0.03 ±0.01 | TTN:-0.003 ±0.008 | MUC2:-0.005 ±0.011 | MUC4:-0.01 ±0.016 | MUC5B:-0.017 ±0.021 | NEFH:-0.136 ±0.009 | FAT1:0.007 ±0.013 | MAML2:-0.054 ±0.02 | MUC16:-0.114 ±0.017 | **OBSCN:0.13 ±0.015** |
| BLCA | TTN:-0.051 ±0.01 | TP53:-0.03 ±0.011 | MUC16:-0.003 ±0.009 | ARID1A:0.008 ±0.013 | KDM6A:-0.05 ±0.013 | SYNE1:-0.055 ±0.011 | PIK3CA:-0.005 ±0.012 | HMCN1:-0.023 ±0.011 | FLG:-0.036 ±0.019 | **CSMD3:0.111 ±0.007** |

with the development of lung cancer [38], which is in line with our prediction that its causal effect on DNA replication (0.006±0.008, in Table 1) and EMT process (0.042±0.005, in Table 2) is not high. *VHL* with the highest mutation frequency in KIRC, but the prognosis of KIRC patients with *VHL* as the therapeutic target is not satisfactory [39,40], which is consistent with the low ATE on DNA replication (0.009±0.004) and EMT process (-0.012±0.004).

The results also suggest that most genes with high mutations don't promote DNA proliferation and EMT process. Genes with ATE values which are infinitely close to 0 or negative indicate that their mutations don't promote or benefit the biological process, as shown in Tables 1 & 2 where a portion of mutations have negative ATE values. Compared with the ATE values listed in Figs 3 and 4, the overall ATE values in Tables 1 & 2 are much smaller. Besides, some mutations may gain potential functions which may even inhibit the activity of cancer-related processes. Taking *NRAS* and *HRAS* for instance, they are members of oncogenic RAS and the clinical impact of RAS mutations on the management of thyroid nodules with indeterminate cytology is unsatisfactory [41,42]. We compare the average expression value of these genes in mutated and non-mutated groups in THCA dataset, and find the value of the mutated group is slightly higher than the non-mutated group (*NRAS*: 866 in the non-mutated group and 880 in the mutated group; *HRAS*: 570 in the non-mutated group and 673 in the mutated group).

## 2.4 Identified key mutations significantly change the activities of cell proliferation and EMT

To statistically illustrate the reliability of our results, we perform the Mann-Whitney U test (M.W.) [43] to compare the activity differences of cancer-related biological processes, including DNA replication and EMT process, between the mutated and non-mutated groups of mutations with the highest ATE value across 10 cancers. As shown in Figs 5 and 6, the activity

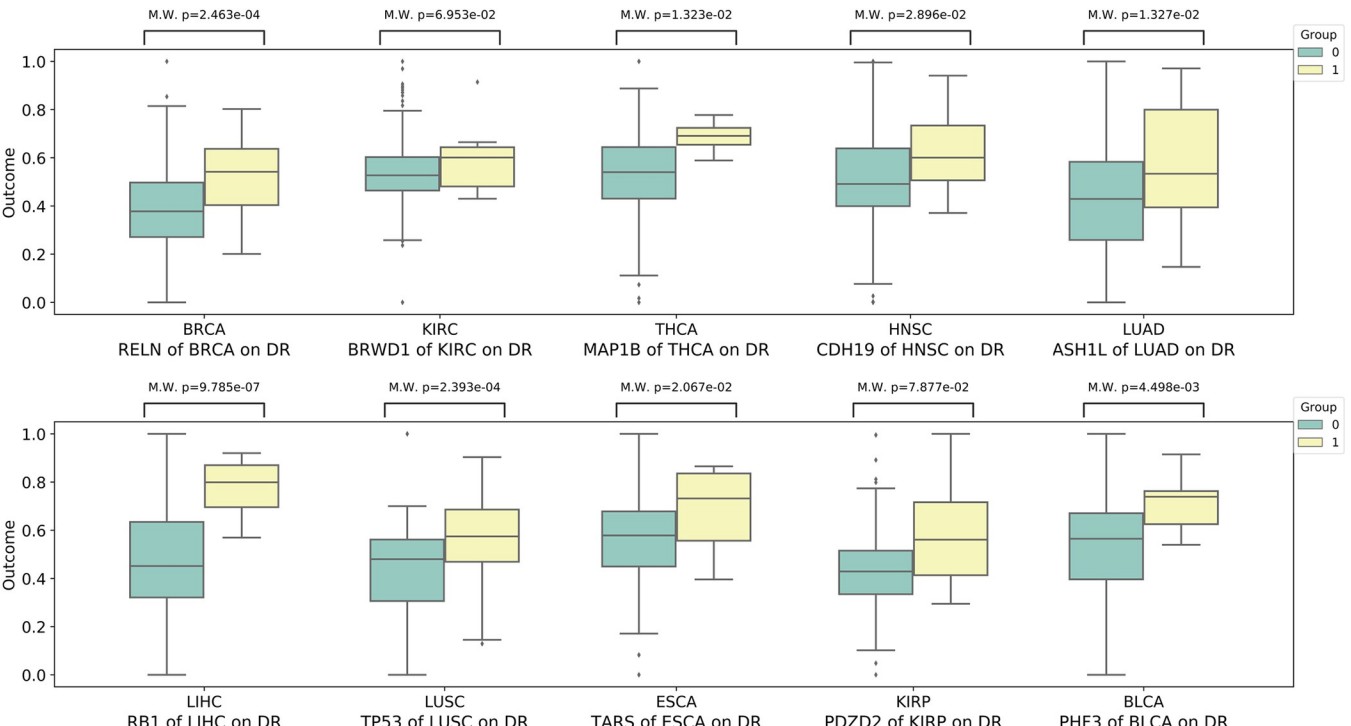

**Fig 5. The Mann-Whitney U test (M.W.) analysis reflects the significant differences between mutated groups and non-mutated groups of mutated genes with high ATE values in the activity of DNA replication (DR) for 10 cancers.**

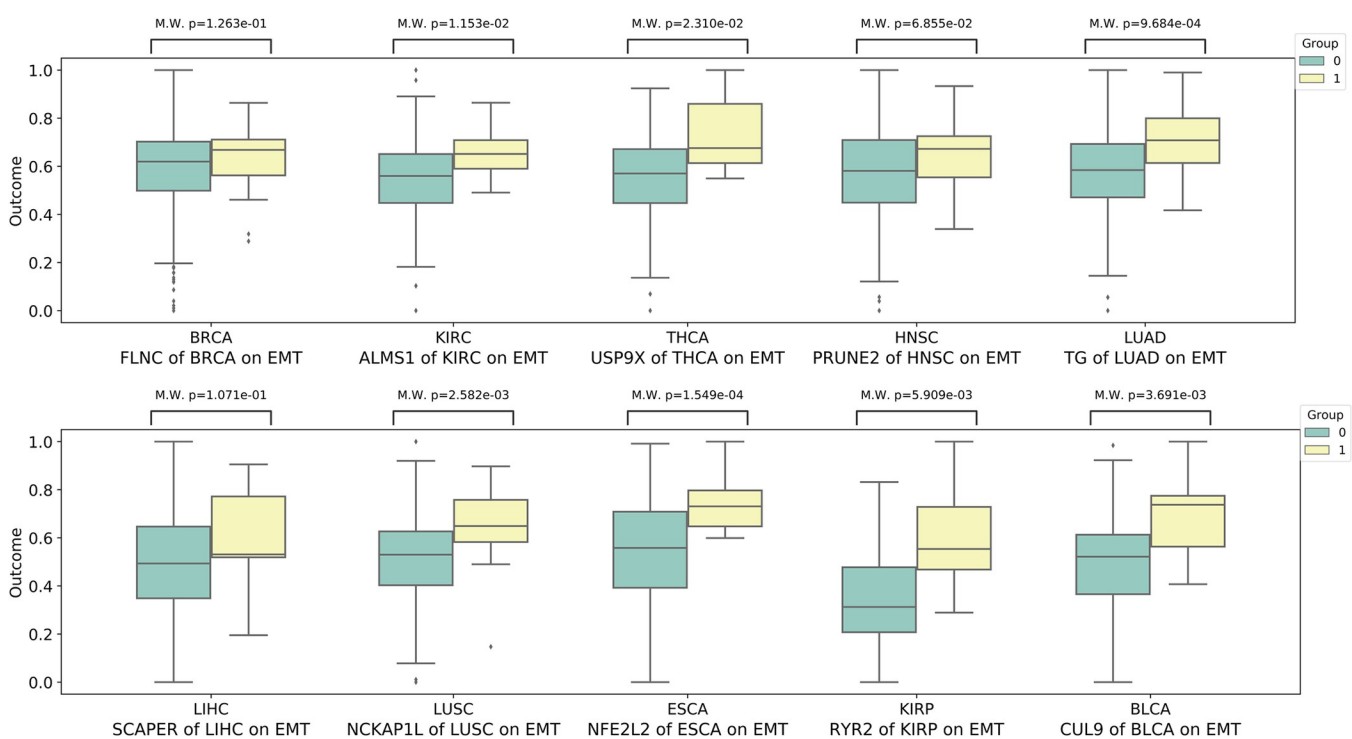

**Fig 6. The Mann-Whitney U test (M.W.) analysis reflects the significant differences between mutated groups and non-mutated groups of mutated genes with high ATE values in the activity of EMT process for 10 cancers.**

of both DNA replication and EMT process of mutant genes with high ATE values identified by CEBP in mutated groups and non-mutated groups are significantly different (p-value $< 0.05$) in 10 cancers. Although a small proportion of these key mutations have p-values above 0.05, their promotion effects on cancer biological processes are strongly supported by existing literature and studies. For instance, *FLNC*, which is identified as the key mutation of EMT process of BRCA, is crucial in cell contraction and spreading as a large actin-cross-linking protein [44,45] and is important in the lymph node metastasis of cancers [46]. Aberrant expression and methylation of *PRUNE2* are reportedly associated with EMT process and nodal metastases in head and neck cancer (HNSC) [47].

In addition, we also conduct M.W. analysis between the mutated and non-mutated groups of genes with the highest mutation rates in 10 cancers. As shown in Figs 7 and 8, except for a few mutations, most listed mutations do not have significant activity differences between these two groups, such as *VHL* on DNA replication and EMT in KIRC, suggesting there is no significant correlation between the change of the biological process activity and the gene mutation frequency. In addition, we find that a mutation may have dual effects on the activity of different biological processes. For example, *TP53* significantly promotes the DNA replication in LUSC but acts as a suppressor of the EMT process. Such effects of mutations on different biological processes may be the reason why the current targeted therapy is not as effective as expected.

## 3 Method

### 3.1 Data collection and preprocessing

We collect genomics and transcriptomics data across 10 cancer types from TCGA [48] and retain the tumor samples which have both mutation and RNA-seq data profiles. Non-

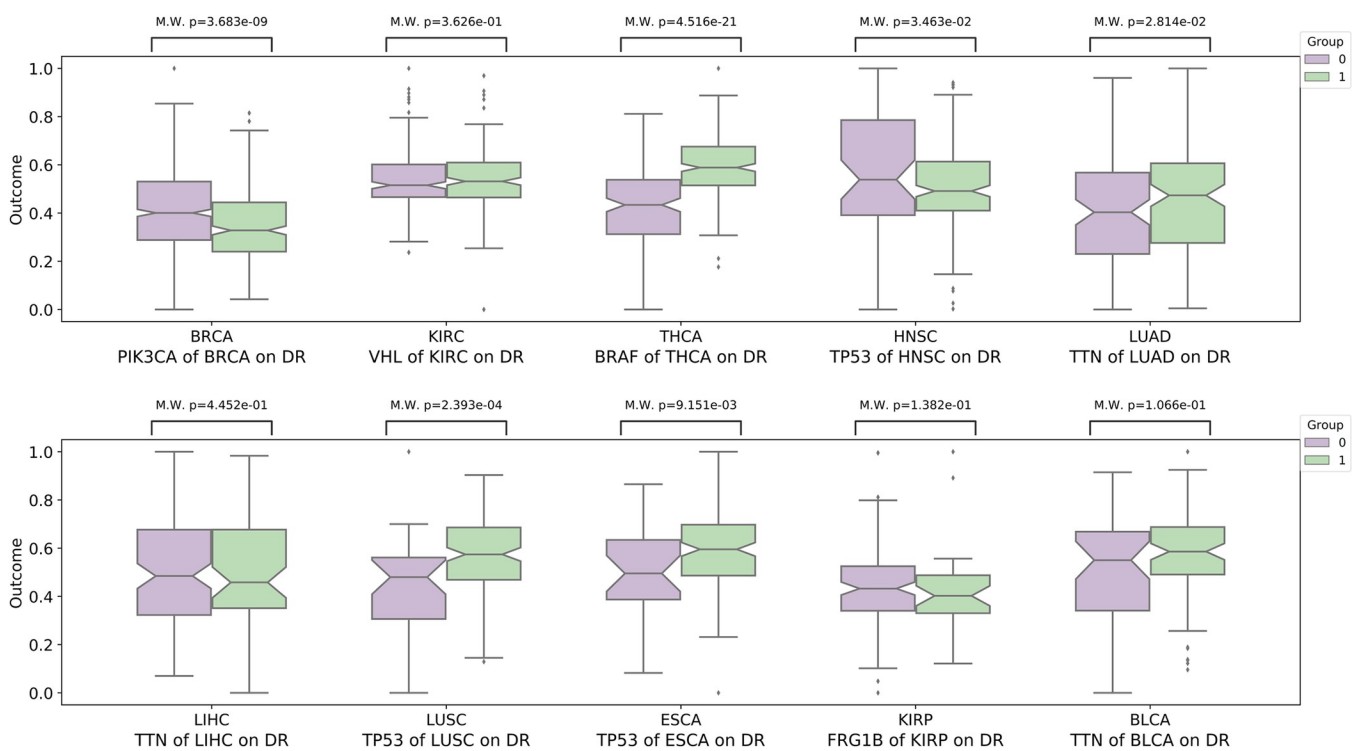

**Fig 7. The Mann-Whitney U test (M.W.) analysis shows there is no significant differences between mutated groups and non-mutated groups of mutated genes with high mutation rates in the activity of DNA replication (DR) for 10 cancers.**

expressed genes (FPKM value less than 10) are removed from the data. For each cancer, the number of samples and the number of genes with a mutation frequency of more than 7% and 3% for cancers are very different, as shown in Table 3. Therefore, in the experiments of Section 2.2, we set different mutation rates or number thresholds for different cancers to identify the key mutations. The gene sets of DNA replication and EMT pathway are downloaded from Gene Set Enrichment Analysis dataset [49].

In general, genes with high mutation rates are more likely to be key mutations and affect the biological processes. So we rank the gene according to the mutation rate from high to low, preferentially select the high-ranking gene as the confounder, and ensure mutation rates of confounders and treatment variables are not less than 1% in all experiments.

## 3.2 Estimating biological processes' activity of cancer samples by regression analysis

As there is no direct biological processes activity reported in the data, we propose a reliable method to quantify the relative biological processes activity of each sample by performing regression analysis on core genes. For a biological processes or pathway consisting of $P$ genes, we generate the transcriptome gene expression matrix of N samples of the pathway denoted by $U = (u_1, \ldots, u_N)^T = (g_1, \ldots, g_P) \in \mathbb{R}^{N \times P}$, where $u_i = (u_i^1, \ldots, u_i^P)^T \in \mathbb{R}^P$ is the i-th sample vector and $g_j = (u_1^j, \ldots, u_N^j) \in \mathbb{R}^N$ is the j-th gene vector. With the matrix $U$, we calculate the Pearson correlation between $P$ genes and obtain two real symmetric matrices $R = (r_i^j) \in \mathbb{R}^{P \times P}$ and $S = (s_i^j) \in \mathbb{R}^{P \times P}$, where $r_i^j$ is the Pearson correlation coefficient between i-th and j-th genes and $s_i^j$ is the corresponding p-value.

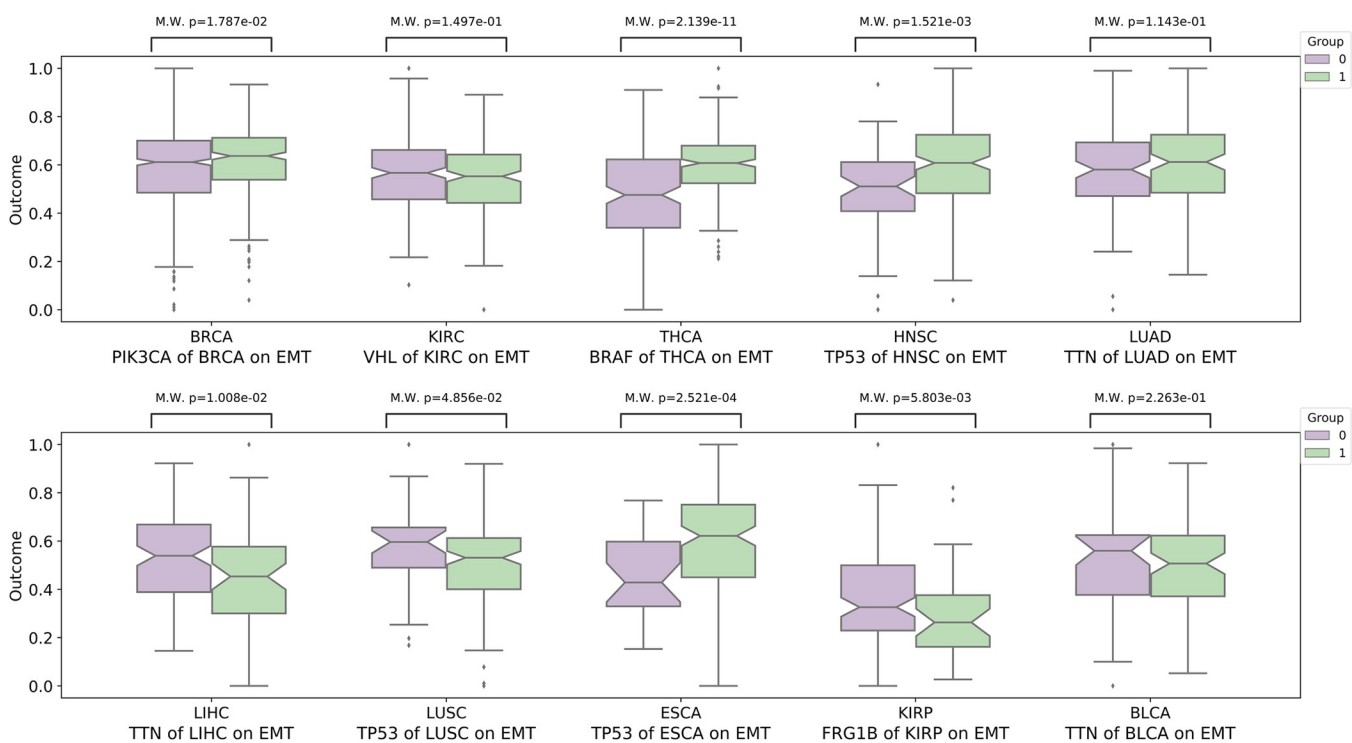

**Fig 8. The Mann-Whitney U test (M.W.) analysis shows there is no significant differences between mutated groups and non-mutated groups of mutated genes with high mutation rates in the activity of EMT process for 10 cancers.**

For j-th gene, we get the content of information interact with other genes labeled as $g_j^{scor}$ and the average expression $g_j^{exp}$ by:

$$g_j^{scor} = \sum_{i=1}^{P} r_i^j \times \chi_{(s_i^j < \alpha)}, \chi_{(s_i^j < \alpha)} = \begin{cases} 1 \text{ if } s_i^j < \alpha \\ 0 \text{ if } s_i^j \geq \alpha \end{cases}, g_j^{exp} = \frac{1}{N} \sum_{i=1}^{N} u_i^j \qquad (2)$$

**Table 3. The statistics of the cancer datasets used in this study.**

| Cancer | #Samples | #Genes with more than 7% of the mutation rate | #Genes with more than 3% of the mutation rate |
|---|---|---|---|
| Breast invasive carcinoma (BRCA) | 949 | 8 | 58 |
| Kidney renal clear cell carcinoma (KIRC) | 415 | 6 | 38 |
| Thyroid carcinoma (THCA) | 396 | 2 | 6 |
| Head and Neck squamous cell carcinoma (HNSC) | 240 | 71 | 605 |
| Lung adenocarcinoma (LUAD) | 230 | 353 | 1905 |
| Liver hepatocellular carcinoma (LIHC) | 178 | 40 | 400 |
| Lung squamous cell carcinoma (LUSC) | 176 | 401 | 2091 |
| Esophageal carcinoma (ESCA) | 162 | 66 | 714 |
| Kidney renal papillary cell carcinoma (KIRP) | 150 | 20 | 212 |
| Bladder Urothelial Carcinoma (BLCA) | 125 | 293 | 2494 |

where $\chi_{()}$ is the indicator function, $\alpha$ is a predefined hyperparameter to drop the correlation parameter with low significance. Then we choose half the number of the genes with higher $g^{scor}$ value which can be considered as core ones denoted as $g_{j^*}$, form the corresponding features average expression vector $K = (g_{1^*}^{exp}, \cdots, g_{[\frac{p}{2}]^*}^{exp}) \in \mathbb{R}^{N \times [\frac{p}{2}]}$ on $N$ samples and the new data matrix of N items sample $U^* = (u_1^*, \cdots, u_N^*)^T = (g_{1^*}, \cdots, g_{[\frac{p}{2}]^*}) \in \mathbb{R}^{N \times [\frac{p}{2}]}$, where [] is floor function. Finally, we calculate the linear regression coefficient of each sample $u_i^* = (u_i^{1^*}, \cdots, u_i^{[\frac{p}{2}]^*})^T \in \mathbb{R}^{[\frac{p}{2}]}$ on $K$ labeled as $y_i$ by:

$$y_i = \min_{y_i} \|y_i u_i^* - K\|_2 \tag{3}$$

With the regression coefficients vector $Y \in \mathbb{R}^N$ of $N$ sample and somatic mutation data (X, M) as observational data, we can calculate the ATE of a mutation on a biological process.

## 3.3 Estimating causal effect based on variational autoencoder model

Given the complex non-linear and high-dimension characters of the biological system, we consider a deep neural network to learn the latent-variable causal model called Causal Effect Variational Autoencoder [30] and extend it to this study. The model consists of a inference network as the encoder and a model network as the decoder.

Inference network intents to learn the posterior representation $q(z|m,x,y)$ with the input data $(m,x,y)$. The distribution of hidden variable Z is assumed as a multivariate Gaussian distribution and estimated means and variances by a two-stage neural network. The first stage is a shared-layer neural network to learn the representation $g(x_i,y_i)$; in the second stage, the inference network is divided into two branch: samples with gene mutated are trained by the $f_1$ neural network, and samples with gene non-mutated are trained by the $f_0$ neural network, where $f$ denotes the multi-layer neural network:

$$q(z_i|m = 0, x_i, y_i) = \prod_{j=1} \mathcal{N}(\mu_j, \varepsilon_j^2); \ \mu_j, \varepsilon_j = f_0(g(x_i, y_i))$$

$$q(z_i|m = 1, x_i, y_i) = \prod_{j=1} \mathcal{N}(\mu_j, \varepsilon_j^2); \ \mu_j, \varepsilon_j = f_1(g(x_i, y_i)) \tag{4}$$

In the Model network, we reconstruct the data $(x,m,y)$ from the hidden variable $z$ : $p(x_i, y_i, m_i|z_i) = p(x_i|z_i)p(m_i|z_i)p(y_i|z_i, m_i)$ according to Fig 1 and each factor is described as:

$$p(x_i|z_i) = f_x(z_i); p(m_i|z_i) = \text{Ber}(\text{elu}(f_m(z_i)));$$

$$p(y_i|z_i, m_i) = \mathcal{N}(m_i f_{y_1}(z_i) + (1 - m_i)f_{y_0}(z_i), \varepsilon^2), \tag{5}$$

where the prior distribution of $z$ is assumed as the standard normal distribution $\prod_{j=1} \mathcal{N}(z_{ij}|0, 1)$ on each dimension, elu() is the ELU-layer [50] to capture the non-linear representation and the Bernoulli distribution is used for calculating the probability of taking treatment $m_i$. As the activity of the biological process is continuous, the distribution of $y_i$ is parametrized as Gaussian and the mean is also split for each treatment group similar to the inference network and the variance is fixed to $\varepsilon$.

Similar to VAE [51], the model is trained by minimizing the KL divergence between data and reconstruction:

$$\mathscr{L} = \sum_{i=1}^{N} \mathbb{E}_{q(z_i|x_i,m_i,y_i)}[log\, p(x_i, m_i|z_i) + log\, p(y_i|m_i, z_i) + log\, p(z_i) - log q(z_i|x_i, m_i, y_i)] \quad (6)$$

With the optimized model, the ATE can be calculated through Eq 1 of each group from posterior $q(Z|X)$.

### 3.4 Implementation

The algorithm of biological process activity estimation is implemented in R and the variational autoencoder model is implemented in Python based on the CEVAE model using Tensorflow [52]. For the active cancer-related biological process, most genes work together and more than half of genes are significantly co-expressed, as the correlation analysis shown in Fig B of S1 Text. So We set the low significance cutoff of correlations as $10^{-3}$ and set half of the genes in the pathway with higher overall correlation values as core genes.

We use 3-layer, 200-width, ELU [50] non-linearity neural networks to approximate $q(y|x, m)$ and $q(z|m,x,y)$ of the inference network, $p(x|z)$ and $p(y|z,m)$ of the model network. We use single layer, 200-width, ELU non-linearity neural networks to learn $q(m|x)$ of the inference network and $p(m|z)$ of the model network. As the number of samples and gene mutation frequencies are very different across cancer types, we consider these two numbers and make a generic assumption about the number of observed confounders, that is the number is 200. Through conducting a parameter analysis of the hidden variable $z$ (the dimension is set to 10, 20, 30, 40, and 50), we find that the results are stable and are less affected by this parameter as shown in Tables A and B of S1 Text. So we set the dimension of $z$ to 20. The weight decay of all parameters is $10^{-4}$ and the optimizer is Adam [53] with a learning rate of $10^{-3}$. Each cancer dataset is divided into train/validation/testing with 70%/10%/20% splits.

## Discussion

Identification of the key mutations for a cancer-related biological process is essential in cancer research. Existing models usually focus on the association not the causality between mutations and cancer biological processes, which may bring bias due to existence of confounders and make it difficult to directly apply to drug targets and practical clinical applications.

Regarding causal inference, modelling confounders and then eliminating confounders bias is the fundamental challenge because confounders distort the causal effect of the treatment on the outcome. The confounders include observed ones from observations and unobserved hidden ones. In this study, we develop a confounder-free computational framework CEBP to estimate the causal effect of a mutation on a biological process and recommend mutations with high causal effect as key mutations. The key mutations identified by CEBP are mostly consistent with existing studies and literatures. Besides, we also discover new key mutations which promote cancer proliferation or EMT processes for each cancer type. The experimental results indicate that there is no significant correlation between the gene mutations rate and their ability in promoting the activity of cancer-related processes. A proportion of mutations are predicted with negative causal effects and we believe that such mutations don't promote biological processes. It may need further studies based on wet experiments for validation.

High-dimensional confounders lead to a higher chance to satisfy the unconfoundedness assumption and a higher probability to violate the positivity assumption. It is the next step of our research to trade-off between the two important assumptions to introduce more confounders in our models without violating the positivity. As there is not abundant relevant

databases to support the validation of key mutations, we compare CEBP with the state-of-the-art methods for driver mutation identification. When comparing with DriverML [13], Driver-Net [16], ActiveDriver [15], OncoDriveFM [54], Simon [55], and SCS [56], the performance CEBP is comparable to these methods, as shown in Fig A of S1 Text. CEBP also provides new potential key mutations and wet experiments will be conducted to validate our results in future works.

In addition to the 10 cancers and 2 pathways considered in this paper, our approach can also extend to other cancers, more biological processes, and more types of regulators. Cancer is the result of a complex system's breakdown and is usually due to a mutation or a set of mutations leading to uncontrolled growth. In this paper, we only consider the causal effect of single gene mutation. In a further study, we will develop a de-confounding model to estimate the bundling effect of candidate mutation sets. As the number of mutation sets is very large due to different combinations and results in a significant increment of computational cost, combinatorial optimization is also an issue to be addressed.

## Supporting information

**S1 Text.  Section A. Comparison with other methods for discovering single driver mutation. Section B. Parameters analysis of CEBP. Section C. Selection of core genes in CEBP. Table A in S1 Text.** The causal effect of TP53 mutation on DNA replication and EMT of BRCA under different numbers of hidden confounders settings. **Table B in S1 Text.** The causal effect of TP53 mutation on DNA replication and EMT of LUAD under different numbers of hidden confounders settings. **Fig A in S1 Text.** Fraction of predicted driver genes presents in CGC which consists 616 cancer-related mutations. **Fig B in S1 Text.** Heat maps of correlations between genes of DNA replication and EMT in BRCA, LUAD, and LIHC tumor samples, where the color indicates the correlation value and the number on the horizontal and vertical axes is the gene's index.
(DOCX)

## Acknowledgments

We thank Qiang Huang for his comments.

## Author Contributions

**Conceptualization:** Yijun Liu, Ji Sun, Huiyan Sun, Yi Chang.

**Data curation:** Yijun Liu.

**Formal analysis:** Yijun Liu, Ji Sun, Huiyan Sun, Yi Chang.

**Investigation:** Yijun Liu.

**Methodology:** Yijun Liu, Ji Sun, Huiyan Sun, Yi Chang.

**Software:** Yijun Liu.

**Validation:** Yijun Liu.

**Visualization:** Yijun Liu.

**Writing – original draft:** Yijun Liu, Ji Sun, Huiyan Sun, Yi Chang.

**Writing – review & editing:** Yijun Liu, Ji Sun, Huiyan Sun, Yi Chang.

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
