## [Decision Letter · Decision Letter 0]

14 Jun 2022

Dear Miss Sun,

Thank you very much for submitting your manuscript "Identification of essential somatic oncogenic mutation based on a confounder-free causal inference model" for consideration at PLOS Computational Biology.

As with all papers reviewed by the journal, your manuscript was reviewed by members of the editorial board and by several independent reviewers. In light of the reviews (below this email), we would like to invite the resubmission of a significantly-revised version that takes into account the reviewers' comments.

We cannot make any decision about publication until we have seen the revised manuscript and your response to the reviewers' comments. Your revised manuscript is also likely to be sent to reviewers for further evaluation.

Sincerely,

Serdar Bozdag, Ph.D.

Guest Editor

PLOS Computational Biology

Ilya Ioshikhes

Deputy Editor

PLOS Computational Biology

Reviewer's Responses to Questions

**Comments to the Authors:**

Reviewer #1: In this manuscript, the authors propose a novel causal inference model based on a deep variational autoencoder to identify key oncogenic somatic mutations. Different from current key mutation identification methods which focus on the correlation between genetic mutations and their functional impacts, the proposed CEBP method aims at detecting their pure causal relationships by eliminating all the observed and latent confounding bias. It provides a new strategy to solve such related problems. Furthermore, when applying CEBP to 10 cancer types, the results indicate that it has no consistent patterns of the relationship between mutation rate of a gene and its ability to drive cancer, which offers some enlightenment to related studies.

However, from my personal point of view, this manuscript should be improved as followings:

1. The authors should give the design philosophy of quantifying the activity of cancer related biological processes. The current description, from line 133 to 137, is not easy to understand.

2. The authors set 200 observed confounders and 20 hidden confounders in the VAE model. How do these two values be determined?

3. Regarding the effectiveness of CEBP, it is suggested to compare it with relevant gold standard, and verify the validity of conclusions through wet lab experiments or other credible computational experiments.

4. The authors should clarify what the meanings of “measurement of the biological process” in line 129, and “the biological process estimation” in line 132 are. In Section 2.3, the authors should clarify what the meanings of “ a value less than 0 says the treatment can only hurt outcomes and cannot help them” and “the mutation with such ATE is detrimental to biological processes” are.

5. The source code of CEBP should be provided.

6. Please give More description to Figure 2.

7. Reference 31 is not correct.

8. The writing should be greatly improved. Some typos should be corrected, such as “biological procession”, “Fig. 3 & & 4 ”, “A a value of ATE equal to 0 ”, etc.

Reviewer #2: To detect mutational genes capable of driving cancer is a critical issue in cancer research. Existing methods focus on the association between genetic mutations and functional changes in relevant biological processes without considering the causality. In this manuscript, the authors construct a causal inference model based on a deep variational autoencoder to identify key oncogenic somatic mutations. Based on the genomics and transcriptomics data from 10 cancers, the experimental results indicate that genes with higher mutation frequency do not necessarily mean they are more potent in inducing cancer and promoting cancer development. However, despite the exciting idea of the causal inference, this manuscript still has the following issues to be addressed.

Major issue:

1. There are no data and codes on the GitHub page.

2. According to the descriptions in section 2.1, X is a 0-1 matrix, M is a 0-1 vector, and is dataset D only used to calculate Y?

3. Equation 1 should be expanded to explain how those expectations are calculated.

4. In section 2.3, the number of observed and hidden confounders is set up to 200 and 20, respectively. Additional explanations and experiments are required to explain the influence of parameters on the performance.

5. In sections 2.2 and 2.3, a value of ATE equal to 0 means there is no difference in the outcome of the control and treatment group, i.e. a value of ATE not equal to 0 means there is a difference in the outcome. Positive means beneficial, and negative means harmful. So why only consider the positive ATE value, not the absolute value as the causal effect of genes? In this way, the result will be very different from figure 3.

6. Figure 5 only shows the boxplot of TP53, the boxplot of other genes should be included in supplementary materials.

7. In section3.1, the data processing process is described vaguely and should be described more clearly. Make every step in the preprocessing clear. What software was used?

8. In equation 2, the meaning of function � should be further explained.

9. In section 3.2, why choose half the number of the features with higher gcor which can be considered as significant ones?

10. In section 3.3, why choose to set a shared-layer neural network to represent g and unshared-layer neural networks to represent f? What are the benefits of doing this?

11. In equation 4, a detailed explanation of each probability distribution definition should be given.

12. More details should be added about your model training, such as experimental settings and parameter selection.

Minor issues:

1. In figure 2, the “Model based on causal graph” is abstract. The authors should revise figure 2 to make it more intuitive, which would be easier to understand their method procedure.

2. Figure 2 should be captioned.

3. Figure 5 should illustrate that 0/1 group presents mutated or not.

4. Please define the “KL divergence” and give the original citation.

5. There are still some grammar issues. For example, in line 209, it should be “A value”.

Reviewer #3: Please find the attached pdf.

**Have the authors made all data and (if applicable) computational code underlying the findings in their manuscript fully available?**

Reviewer #1: None

Reviewer #2: None

Reviewer #3: None

PLOS authors have the option to publish the peer review history of their article (what does this mean?). If published, this will include your full peer review and any attached files.

Reviewer #1: No

Reviewer #2: **Yes: **Qin Ma

Reviewer #3: No
---

## [Decision Letter · Decision Letter 1]

31 Aug 2022

Dear Miss Sun,

We are pleased to inform you that your manuscript 'Identification of key somatic oncogenic mutation based on a confounder-free causal inference model' has been provisionally accepted for publication in PLOS Computational Biology.

Best regards,

Serdar Bozdag, Ph.D.

Guest Editor

PLOS Computational Biology

Ilya Ioshikhes

Section Editor

PLOS Computational Biology

Reviewer's Responses to Questions

**Comments to the Authors:**

Reviewer #1: It is revised well.

Reviewer #2: All the concerns have been addressed.

**Have the authors made all data and (if applicable) computational code underlying the findings in their manuscript fully available?**

Reviewer #1: Yes

Reviewer #2: Yes

PLOS authors have the option to publish the peer review history of their article (what does this mean?). If published, this will include your full peer review and any attached files.

Reviewer #1: No

Reviewer #2: **Yes: **Qin Ma

---

## [Editor Report · Acceptance letter]

8 Sep 2022

PCOMPBIOL-D-22-00566R1 

Identification of key somatic oncogenic mutation based on a confounder-free causal inference model

Dear Dr Sun,

I am pleased to inform you that your manuscript has been formally accepted for publication in PLOS Computational Biology. Your manuscript is now with our production department and you will be notified of the publication date in due course.

With kind regards,

Zsofi Zombor
